# Using the first nationwide survey on non-communicable disease risk factors and different definitions to evaluate the prevalence of metabolic syndrome in Burkina Faso

Kadari Cissé[1,2]*, Délwendé René Séverin Samadoulougou[3], Joel Dofinissery Bognini[4], Tiga David Kangoye[5], Fati Kirakoya-Samadoulougou[1]

1 Centre de Recherche en Epidémiologie, Biostatistiques et Recherche Clinique, Ecole de Santé Publique, Université libre de Bruxelles, Brussels, Belgium, 2 Departement Biomédical et Santé Publique, Institut de Recherche en Sciences de la Santé, Ouagadougou, Burkina Faso, 3 Service de Médecine, Centre Hospitalier Régional de Banfora, Banfora, Burkina Faso, 4 Centre National de la Recherche Scientifique et Technologique, Unité de Recherche Clinique de Nanoro, Institut de Recherche en Sciences de la Santé, Ouagadougou, Burkina Faso, 5 Institut national de santé publique (INSP), CNRFP, Ouagadougou, Burkina Faso

* cisskad4@gmail.com

**Data Availability Statement:** The dataset of the STEPS survey that was used in this research is

## Abstract

### Background

The burden of cardiovascular diseases is rising in the developing world including Sub-Saharan Africa. The rapid rise of cardiovascular disease burden is in part due to undetected and uncontrolled cardiovascular risk factors. The clustering of metabolic syndrome (MetS) components is associated with a high risk of cardiovascular diseases. This complex biochemical disorder is still poorly studied in western Africa. In this study, we aimed to determine the prevalence of metabolic syndrome and its determinants among the adult population in Burkina Faso.

### Methods

We performed a secondary analysis of the data from the first national survey on non-communicable diseases risk factors using the World Health Organization (WHO) Stepwise approach. We included 4019 study participants aged 25 to 64 years. The metabolic syndrome prevalence was estimated using six different definitions.

### Results

The mean age was 38.6±11.1 years. Women represented 52.4% and three quarters (75%) lived in rural areas. The overall prevalence of metabolic syndrome according to the different definitions was 1.6% (95%CI:1.1–2.2) for the American College of Endocrinology, 1.8% (95%CI: 1.3–2.4) for the WHO, 4.3% (95%CI:3.5–5.2) for the National Cholesterol Education Program Adult Treatment Panel III, 6.2% (95%CI: 5.1–7.6) for the AAH/NHBI, 9.6%

available at the Ministry of Health upon request since the data are owned by a third-party organization (ministry of health). The restriction was imposed by the Ministry of Health. Individuals interested can access to the STEPS survey dataset by emailing the ministry of health (Tel: (+226) 25 25 25 25; email: contact@sante.gov.bf) or directly Dr Paulin Somda: k_admos@yahoo.fr (NCDs program coordinator) or Zoma Torez: torezo2000@yahoo.fr (STEP survey data manager)). All survey materials of STEPS survey are available on the WHO website (https://extranet. who.int/ncdsmicrodata/index.php/catalog). The authors confirm that they did not have any special access privileges that others would not have.

**Funding:** This study is funded by Académie de Recherche et d'Enseignement Supérieur (ARES) of Belgium, in the context of research program for development focused on the cardiovascular diseases in Burkina Faso. The program, which is named CARDIOPREV, is achieved by the Institut de Recherche en Sciences de la Santé (IRSS) in Burkina Faso and the Université Libre de Bruxelles in Belgium. The funder had no role in study design, data collection and analysis, decision to publish, or preparation of the manuscript.

**Competing interests:** The authors have declared that no competing interests exist.

**Abbreviations:** ACE, American College of Endocrinology; AHA/NHLBI, American Heart Association and National Heart Lung and Blood Institute; BMI, body mass index; BP, blood pressure; CVD, cardiovascular disease; DM, diabetes mellitus; FG, fasting glucose; HBP, high blood pressure; HDL, high density lipoprotein; IDF, International Diabetes Federation; JIS, Joint Interim Statement; MetS, metabolic syndrome; NCD, noncommunicable disease; NCEP ATP III, National Cholesterol Education Program Adult Treatment Panel III; T2DM, Type 2 diabetes mellitus; WC, waist circumference; WHO, World Health Organization.

(95%CI: 8.1–11.3) for the International Diabetes Federation and 10.9% (95%: 9.2–12.7) for the Joint Interim Statement. The metabolic syndrome components with the highest prevalence were low High density lipoprotein (63.3%), abdominal obesity (22.3%) and hypertension (20.6%). People living in urban areas and those with older age have higher prevalence of metabolic syndrome regardless of the definition used.

## Conclusion

Our findings suggest various levels of prevalence of MetS according to the definition used. Identifying the most appropriate criteria for MetS among the adult population is important to early detect and treat this syndrome and its components at the primary health care level to control the rising burden of cardiovascular diseases in the context of ongoing epidemiological transition in the country.

## Introduction

Non-communicable diseases (NCDs) included cardiovascular diseases (CVD), are rising worldwide and are considered one of the major health challenges of our century [1]. In developing countries including countries in Sub-Saharan Africa (SSA), NCDs are responsible for 82% (17 million) of the global premature deaths [2]. Among risk factors leading to NCDs, some are preventable such as behavioral and metabolic risk factors [1]. Some of these metabolic risk factors are a part of metabolic syndrome (MetS), which ties together type 2 diabetes mellitus and cardiovascular risk factors including insulin resistance, dyslipidemia, hypertension and abdominal obesity [3, 4].

In the SSA countries, the prevalence of the MetS varies between countries depending on the definition criteria considered [5]. In 2019, a systematic review and meta-analysis of studies conducted in several SSA countries estimated the overall prevalence of MetS in 34,324 healthy participants aged ≥16 years old [6]. According to the different diagnostic criteria, the prevalence varied from 11.1% (95%CI: 5.3–18.9, World Health Organisation (WHO) criteria) to 23.9% (95%CI: 16.5–32.3, Joint Interim Statement (JIS) criteria). The prevalence of MetS was higher in women than in men, and higher in (semi-)urban participants than in rural participants [6]. According to different studies, this prevalence in different adult populations also increased with age [7–11].

In Burkina Faso, it was estimated that non-communicable diseases accounted for around 33% of all deaths in 2016 [12]. However, very few studies specifically investigated the MetS burden. In a survey carried out in the general population of two districts of the capital city, the prevalence of MetS was estimated at 7% [13]. Other hospital-based studies found a MetS prevalence of 10% (International Diabetes Federation (IDF) criteria) or 12.3% (National Cholesterol Education Program Adult treatment Panel III (NCEP-ATP III) criteria) in HIV patients on antiretroviral therapy [14], 17.5% in patients with high blood pressure [15] and 48.9% in patients with diabetes [16].

A 25% reduction in the overall mortality from cardiovascular diseases, diabetes and others NCDs by 2025 is the first goal to which governments, including that of Burkina Faso, are committed to achieving in the Global Monitoring Framework for NCDs [17]. The importance of MetS in this context lies in its association with increased cardiovascular mortality and all-cause mortality in the general population [18]. It is therefore of utmost importance to get an

estimate of its extent in the general population of Burkina Faso where data and research supporting decision-making relative to NCDs prevention/control strategies and impact evaluation are very limited [19]. Thus, the objective of our study was to exploit unique population-based survey data on non-communicable diseases in Burkina Faso, to estimate, for the first time, the national prevalence of the metabolic syndrome using different definitions of MetS and its determinants.

## Materials and methods

### Study setting

Burkina Faso is a land-locked country in West Africa with an area of 272,967.47 km$^2$. The population was 20,244,080 inhabitants, and people aged over 25 years represented 36% of the population in 2018 according to the estimation of the National Institute of Statistics and Demography [20]. Life expectancy was 56.7 years [21]. Many risk factors contributing to the occurrence of NCDs were reported, and about 33% of all deaths are due to NCDs. Its climate is tropical with a long dry season making it difficult to grow fruits. About 81% of workers were farmers. Administratively, the country is subdivided into 13 regions, 45 provinces, 350 departments, 351 municipalities and 8,228 villages [20].

### Data source

**a) Study design and population.** Our study used data from a cross-sectional survey, the national WHO Steps survey aiming to assess the risk factors for NCDs which was conducted in the 13 regions of Burkina Faso from 26 September to 18 November 2013. Data were collected from a representative sample of people aged 25–64 years old in Burkina Faso. the study was designed to have estimates at the national, regional and place of residence (urban/rural) level. Participants were identified using a three-stage cluster stratified survey sampling. The sampling frame used was taken from the 2006 general population and housing census [22] and updated in 2010 during the Burkina Faso Demographic and Health Survey [23]. This update concerned the enumeration areas (EAs) that correspond to the cluster within the framework of this study. Clusters were organized using region and place of residence. One respondent was identified in each of 4800 selected household [24].

**b) Data collection.** Data were collected electronically on Personal Digital Assistant (PDA) and consisted of face-to-face interviews after obtaining informed consent from the participant for risk factors linked to NCDs. The data were collected by taking physical and biochemical measurements from the subjects selected to participate in the survey using the WHO STEPS instrument. The survey was conducted in three steps: the first step focused on socio-demographic information, behavioral measures, questions on physical activity, and food hygiene, oral health, screening for cervical cancer and knowledge risk factors for NCDs. Behavioral measures related to the consumption of tobacco and alcohol. The second step measured the following physical parameters: weight, height, waist circumference and blood pressure. At the third step blood sugar and blood cholesterol were measured.

### Variables of interest

**a) Main outcome or dependent variable.** In our study, the dependent variable was a binary variable measure at the individual level that can take two possible values: the presence of metabolic syndrome (1), or absence (0). These criteria were used to define MetS according to the definitions presented in Table 1. These MetS definition were adapted since blood triglyceride level was not collected during the STEP survey in Burkina Faso, due to a lack of adequate

**Table 1. Criteria for metabolic syndrome for each definition.**

| Criteria | WHO (1998–1999) [29, 30] | NCEP-ATPIII (2001) [31] | IDF (2006 [32]) | AACE 2003 [33] | NHLB/AHA [34] | JIS (2009) [35] |
|---|---|---|---|---|---|---|
| **Mandatory criteria** | • DM<br><br>Or<br><br>• FG≥5.6 | (None) | • WC≥ 94cm (M)/ 80cm (W)<br><br>Or<br><br>• BMI > 30 kg/m$^2$ | • ≥ 2 characteristics (non-diabetic patients):<br>• 6.1 ≤ FG< 7.0 mmol/l<br>• HDL-c < 1.03 mmol/l (M) / 1.29 mmol/l (W)<br>• BP> 130/85 mmHg | (None) | (None) |
| **Conditions** | Plus≥ 2 other criteria | ≥ 3 criteria | Plus ≥ 2 other criteria | Plus ≥ 1 other criterion | ≥ 3 criteria | ≥ 3 criteria |
| **Other FG criterion** | | FG≥ 6.1mmol/l | FG≥ 5.6 mmol/l<br><br>Or known T2DM | Family history of T2DM, | FG≥ 5.6 mmol/l<br>or treatment | FG≥ 5.6 mmol/l<br>or treatment |
| **Other obesity criterion** | BMI> 30 kg/m$^2$ | WC>102cm (M)<br><br>WC>88cm (W) | | BMI>25.0 kg/m2; | WC>102cm (M)<br><br>WC>88cm (W) | WC≥ 94cm (M)<br><br>WC≥ 80cm (W) |
| **Other HDL-c (mmol/l) criterion** | HDL-c<br><0.9mmol/l(M)<br><br><1.0mmol/l (W) | HDL-c<br><1.03mmol/l (M)<br><br><1.3mmol/l (W) | HDL-c treatment or<br><1.03mmol/l (M)<br><br><1.29mmol/l (W) | | HDL-c<br><1.03mmol/l (M)<br><br><1.3mmol/l (W) | HDL-c treatment or<br><br>< 1.0 mmol/l (M)<br><br>< 1.3 mmol/l (W) |
| **Other BP criterion** | BP≥ 140/90 mmHg | BP≥130/85 mmHg | BP≥ 130/85 mmHg or treatment | • Known HBP or<br>• Family history of HBP | BP≥130/85 mmHg<br>or treatment | BP≥130/85 mmHg<br>or treatment |
| **Else criterion** | | | | • Non-Caucasian ethnicity<br>• Sedentary lifestyle<br>• Age>40 years | | |

AACE: American College of Endocrinology; AHA/NHLBI: American Heart Association and National Heart Lung and Blood Institute; IDF: International Diabetes Federation; JIS: Joint Interim Statement; NCEP-ATPIII: National Cholesterol Education Program Adult Treatment Panel III; WHO: World Health Organization; DM: diabetes mellitus; T2DM: Type 2 Diabetes mellitus; HDL-c: high density lipoprotein; BMI: body mass index; BP: bloodpPressure; WC: waist circumference; FG: fasting glucose; HBP: high blood pressure.

sampling equipment. The measurement of MetS components was done following the WHO Stepwise approach [25]. Previous studies using the STEP survey had already reported details of the measurement of two MetS components: hypertension [26] and Fasting blood glucose [27]. The level of blood lipid (total cholesterol and high density lipoprotein cholesterol (HDL) was obtained using a portable device (Cardio-Chek P•A™ SILVER), which also provided the level of glucose, using a whole blood sample obtained from a finger prick [27, 28]. The waist circumference was measured with a tape measure, applied directly to the skin [28]. The following variables were used to compute MetS according to each definition (see Table 1 for more details):

- **Anthropometric factor**: weight (Kg), height (m), body mass index (BMI) (18.5≤ BMI <25 = normal, BMI ≥25 = overweight, BMI ≥30 = obesity), and waist circumference, blood pressure

- **Biological factors**: total cholesterol, cholesterol HDL, fasting glycemia,

- **Treatment**: treatment of diabetes (yes/no), the treatment of hypertension (yes/no) and treatment of dyslipidemia (yes/no).

   **b) Explanatory variables.**   Explanatory variables are detailed below:

- Socio-demographic factors: age (25–34, 35–44, 45–54, and 55–64 years), gender (male/female), education level (no level, primary, secondary or higher), place of residence (urban/rural), region, profession (wage earner, self-employed or jobless),

- Behavioral factors: smoking (yes/no), use of alcohol (yes/no), number of fruits or vegetables eaten per day, fat intake (yes/no), and number of meals taken out of the household, physical activities: physical activities (high, moderate and low intensity),

## Statistical method

We first described the characteristics of the study population through a weighted analysis for complex samples. The MetS weighted prevalence was presented according to definitions and sociodemographic characteristics. Chi-square tests were used to assess association of the independent variables with MetS. We implemented a modified Poisson regression model using a generalised estimating equation to derive prevalence ratios (PR) taking into account the clustering of observations. In our analysis strategy, we implemented an analysis with complete case data. PR with a 95% confidence interval were calculated. Statistical significance was accepted at the 5% level (p <0.05).

## Ethics approval and consent to participate

Before collecting data in the field, the survey protocol was approved by the Ministry of Health's Ethics Committee for Health Research, and the Ministry of Scientific Research and Innovation, and informed consent was required before the participation of any individual selected for the investigation (Deliberation No. 2012-12-092 of 05 December 2012). The confidentiality of the information collected has been mentioned in the informed consent form. Our analysis was also approved by the Ministry of Health's Ethics Committee for Health Research (Deliberation No. 2020-10-231 of October, 07, 2020).

## Results

### Sociodemographic profile

A total of 4019 participants have met the inclusion criteria (see Fig 1). The mean age was 38.6 ±11.1 years and 41.2% were younger than 35 years old. Women represented 52.4% of the study population. Three-quarters (75%) lived in rural areas (see Table 2).

### Behavioral and metabolic risk profile

In terms of behavioral risk, 11.3% of the study population were smokers, 4.3% had a history of alcohol abuse, 95.5% had a poor fruit and vegetable intake and 14.0% had low physical activity. The prevalence of obesity was 4.4%. The prevalence of abdominal obesity as defined by JIS was 22.3%. One-fifth (20.6%) had hypertension. Fasting blood glucose levels were high in 5.1% of the study population. High total cholesterol and low HDL have been reported in 2.2% and 63.3% of the study population respectively (see Table 3).

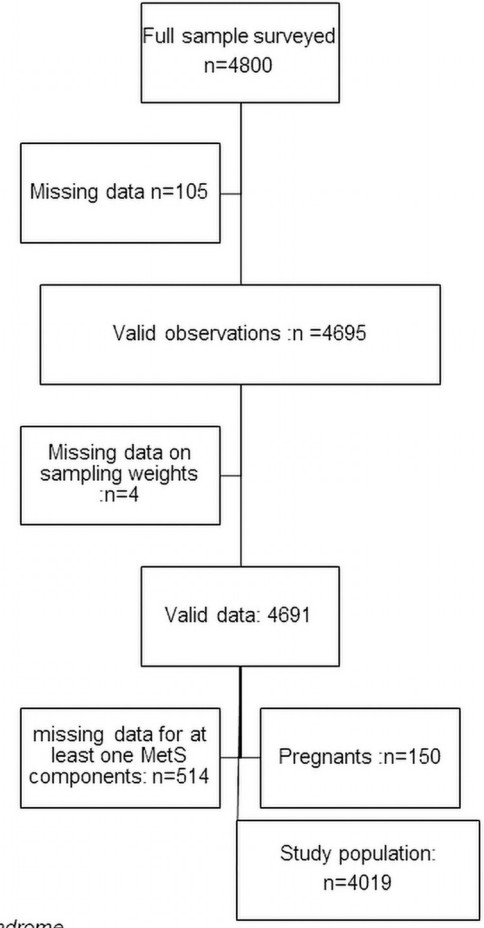

**Fig 1. Diagram flow of study participants.**

## Prevalence of metabolic syndrome (MetS)

The prevalence of MetS varied widely depending on the definition used. The highest prevalence was observed by using the JIS definition (10.9%) followed by IDF (9.6%), American Heart Association and National Heart Lung and Blood Institute (AHA/NHLBI (6.2%) and NCEP-ATP III (4.3%). The lowest prevalence was found using American college of Endocrinology (AACE) (1.6%) and the WHO criteria (1.8%) (Tables 4 and 5). The prevalence of MetS was higher among women compared with men in four out of the six definitions used. Based on the WHO and AACE definitions, there was no gender difference in the prevalence of MetS.

## Determinants of METS

The results of the multivariable analysis are reported in Tables 6 and 7. Among the sociodemographic characteristics listed in Table 2, age and residence were the main determinants of MetS regardless of the definition used. People living in urban areas had a higher risk clustering of MetS components compared with those living in rural areas. The risk of MetS increased with age group, thus participants aged 55 to 64 years had a higher risk compared to those aged 25 to 34 years regardless of the definition used. Gender disparities were reported

**Table 2. Sociodemographic characteristics of the study population.**

| Sociodemographic characteristics | Number | Percentage* |
|---|---|---|
| Age group, years | 4019 | |
| 25–34 | 1781 | 41.2 |
| 35–44 | 1011 | 27.8 |
| 45–54 | 743 | 19.4 |
| 55–64 | 484 | 11.7 |
| Gender | 4019 | |
| Women | 2011 | 52.4 |
| Men | 2008 | 47.6 |
| Marital status | 4014 | |
| Single | 575 | 13.2 |
| Married | 3439 | 86.8 |
| Completed level of education | 4011 | |
| No formal school | 3166 | 79.1 |
| Primary school | 596 | 14.5 |
| Secondary or more | 249 | 6.4 |
| Profession | 4014 | |
| Wage earner | 195 | 5.1 |
| Self-employed | 2875 | 69.0 |
| Jobless | 944 | 26.0 |
| Residence | 4019 | |
| Urban | 787 | 25.0 |
| Rural | 3232 | 75.0 |
| Region | 4019 | |
| Boucle du Mouhoun | 419 | 9.4 |
| Cascades | 133 | 2.7 |
| Centre | 311 | 9.3 |
| Centre-Est | 352 | 7.5 |
| Centre-Nord | 377 | 10.3 |
| Centre-Ouest | 352 | 7.7 |
| Centre-Sud | 198 | 4.4 |
| Est | 345 | 8.4 |
| Haut-Bas | 454 | 12.3 |
| Nord | 383 | 7.7 |
| Plateau-Central | 213 | 7.8 |
| Sahel | 289 | 8.4 |
| Sud-Ouest | 193 | 4.1 |

*Weighted percentage.

for four definitions (NCEP-ATP III, IDF, AHA/NHLBI, JIS). For each of these definitions men had a lower risk of MetS compared to women, however, using the WHO and AACE criteria we did not find any difference between men and women regarding MetS risk. We did not find any association between marital status, profession and MetS under any of the definitions. Finally, according to the WHO, NCEP-ATP III, AACE, and AHA/NHLBI definitions, people who did not receive formal education had a lower risk of MetS compared to those who attended school.

**Table 3. Descriptive behavioral and metabolic characteristics of risk factors among the study population.**

| Risk factors | Number | Weighted percentage |
|---|---|---|
| **Behavioral risk factors** | | |
| Current smoker (n = 4019) | 496 | 11.3 |
| Excessive drinker (n = 4019) | 174 | 4.3 |
| Fruit and vegetable intake | 3731 | |
| <5 | 3545 | 95.5 |
| ≥5 | 186 | 4.5 |
| Type of fat most commonly used | 3912 | |
| Vegetable oil | 2431 | 63.1 |
| Butter, lard or fat, margarine | 1073 | 27.0 |
| None or other | 408 | 9.9 |
| Number of meals taken out of the household | 3987 | |
| <8 | 3890 | 97.8 |
| ≥8 | 97 | 2.3 |
| Physical activity | 4019 | |
| Intense | 2452 | 60.6 |
| Moderate | 1037 | 25.4 |
| Low | 530 | 14.0 |
| **metabolic risk factors** | | |
| BMI class | 4019 | |
| Underweight | 455 | 11.9 |
| Normal | 2898 | 70.5 |
| Overweight | 510 | 13.3 |
| Obese | 156 | 4.4 |
| High blood pressure (n = 4019) | 808 | 20.6 |
| Elevated blood glucose (n = 4019) | 192 | 5.1 |
| Hypercholesterolemia (n = 4019) | 86 | 2.2 |
| Low HDL-Cholesterol (n = 4019) | 2575 | 63.3 |
| Abdominal obesity (4019) | 812 | 22.3 |

## Discussion

To the best of our knowledge, the present study is the first estimation of the burden of MetS and its components in the adult population of Burkina Faso using different definitions of MetS. Our study reported various prevalence of MetS in Burkina Faso, depending on the definition used. Based on the most recent harmonized definition (JIS), we found that 10.9% of the adult population had MetS. We also found that despite the heterogeneity in the definition of MetS, its components were clustered in urban areas and older adults (55 years and more). The differences in the definitions lie mainly in the choice of the thresholds of MetS components [36]. Previous studies reported the great variability of MetS prevalence according to the definitions [37, 38]. Beyond the debate on the appropriate definition of MetS among African populations [39], it is well known that the clustering of MetS significantly increase the risk of many chronic diseases including CVD, diabetes and cancer [38, 40]. The triglyceride level was not considered in the definition of MetS in our study since it was not collected during the STEP survey in Burkina Faso. This might contribute to lower the prevalence of MetS in this study. Nevertheless, the clustering components of MetS doubled the risk of death and tripled the risk of CVD compared to people without the MetS [41]. Because of this, it is important to engage in ongoing efforts (national plan to address NCD and its risk factors including MetS

**Table 4. Prevalence of metabolic syndrome according to each of six definitions by sociodemographic characteristics.**

| Sociodemographic characteristics | WHO | | NCEP-ATP III | | IDF | |
|---|---|---|---|---|---|---|
| | n | % [95% CI] | n | % [95% CI] | n | % [95% CI] |
| All participants | 68 | 1.8 [1.3–2.4] | 165 | 4.3 [3.5–5.2] | 344 | 9.6 [8.1–11.3] |
| **Age group, years** | | | | | | |
| 25–34 | 17 | 1.0 [0.6–1.6] | 48 | 2.9 [2.1–4.0] | 101 | 6.2 [4.9–7.8] |
| 35–44 | 19 | 2.0 [1.1–3.4] | 40 | 4.3 [2.9–6.3] | 88 | 9.4 [7.2–12.4] |
| 45–54 | 17 | 2.2 [1.2–4.0] | 41 | 5.3 [3.7–7.5] | 89 | 13.9 [10.5–18.1] |
| 55–64 | 15 | 3.9 [2.1–7.4] | 36 | 7.5 [5.0–11.1] | 66 | 14.9 [11.3–19.4] |
| **Gender** | | | | | | |
| Women | 34 | 1.8 [1.2–2.7] | 124 | 6.2 [4.9–7.8] | 278 | 14.7 [12.3–17.5] |
| Men | 34 | 1.9 [1.2–2.9] | 41 | 2.2 [1.5–3.1] | 66 | 4.0 [3.0–5.2] |
| **Marital status** | | | | | | |
| Single | 11 | 2.8 [1.4–5.5] | 30 | 6.3 [4.1–9.5] | 59 | 11.5 [8.4–15.7] |
| Married | 57 | 1.7 [1.2–2.4] | 134 | 4.0 [3.2–4.9] | 283 | 9.3 [7.8–11.0] |
| **Completed level of education** | | | | | | |
| No formal school | 42 | 1.2 [0.8–1.7] | 109 | 3.3 [2.6–4.2] | 251 | 8.6 [7.2–10.2] |
| Primary school | 13 | 2.9 [1.5–5.6] | 38 | 7.9 [5.7–10.8] | 57 | 12.6 [9.2–17.1] |
| Secondary or more | 13 | 7.1 [3.5–13.9] | 18 | 7.6 [4.2–13.3] | 35 | 15.0 [10.3–21.2] |
| **Profession** | | | | | | |
| Wage earner | 7 | 3.4 [1.4–7.7] | 13 | 5.6 [3.1–9.9] | 22 | 11.8 [7.7–17.5] |
| Self-employed | 39 | 1.5 [1.0–2.3] | 93 | 3.4 [2.6–4.4] | 187 | 7.3 [5.8–9.2] |
| Jobless | 22 | 2.3 [1.4–3.8] | 58 | 6.3 [4.5–8.8] | 132 | 15.1 [12.2–18.5] |
| **Residence** | | | | | | |
| Urban | 29 | 4.2 [2.7–6.5] | 80 | 9.9 [7.2–13.5] | 154 | 20.7 [15.8–26.6] |
| Rural | 39 | 1.1 [0.7–1.5] | 85 | 2.4 [1.9–3.0] | 190 | 5.9 [4.9–7.1] |

WHO: World Health Organization; NCEP-ATPIII: National Cholesterol Education Program Adult Treatment Panel III; IDF: International Diabetes Federation, CI: Confidence interval; n = number of participants with metabolic syndrome.

components [42] and the training of physicians specialized in the treatment of metabolic disorders [43]) for early detection and treatment of MetS and its components to reduce CVD.

A previous population-based study in an urban area (Ouagadougou) using the JIS definition of MetS, reported a prevalence of 10.3% in 2012, which is similar to our findings using the same definition [44]. The prevalence of MetS reported in our study is seem to be lower than those reported elsewhere in Africa regardless of definition used. Indeed, a meta-analysis achieved in 2019, found that the pooled prevalence of MetS in the SSA adult population according to different definitions was 11.1% (WHO), 17.1% (NCEP-ATP III), 18.0% (IDF) and 23.9% (JIS) [45]. This meta-analysis also showed that the prevalence of MetS was lower in western Africa compared to southern and eastern Africa. The reasons for this lower prevalence are unclear and thus require further investigations. Yet, adults with any components of MetS and those classified as having MetS have a significantly higher risk of CVD, which justifies the need to identify adults with a clustering of MetS components in routine practice in low and middle-income countries such as Burkina Faso [46]. In these countries, especially in SSA, the rapid rise in morbidity and mortality from CVD has been linked to undetected and uncontrolled vascular risk factors [47].

The components of MetS with the highest prevalence in this study were low HDL, abdominal obesity and hypertension, which is in accordance with those reported in other studies in Africa. Indeed, the main components of MetS among Africans are low HDL, obesity, and

**Table 5. Prevalence of metabolic syndrome according to each of six definitions by sociodemographic characteristics.**

| Sociodemographic characteristics | AACE | | AHA/NHLBI | | JIS | |
|---|---|---|---|---|---|---|
| | n | % [95%CI] | n | % [95%CI] | N | % [95%CI] |
| All participants | 60 | 1.6 [1.1–2.2] | 227 | 6.2 [5.1–7.6] | 390 | 10.9 [9.2–12.7] |
| **Age group, years** | | | | | | |
| 25–34 | 19 | 0.8 [0.5–1.4] | 64 | 4.0 [3.0–5.3] | 116 | 7.0 [5.6–8.7] |
| 35–44 | 13 | 1.6 [0.9–2.8] | 60 | 6.8 [4.8–9.4] | 99 | 10.4 [8.1–13.4] |
| 45–54 | 15 | 2.3 [1.3–4.1] | 56 | 7.6 [5.4–10.6] | 98 | 15.4 [11.5–20.4] |
| 55–64 | 13 | 3.3 [1.6–5.6] | 47 | 10.6 [7.7–14.6] | 77 | 17.8 [13.7–22.7] |
| **Gender** | | | | | | |
| Women | 32 | 1.7 [1.1–2.4] | 168 | 9.0 [7.2–11.1] | 295 | 15.8 [13.2–18.7] |
| Men | 28 | 1.5 [0.9–2.4] | 59 | 3.2 [2.3–4.4] | 95 | 5.5 [4.4–6.8] |
| **Marital status** | | | | | | |
| Single | 12 | 1.7 [0.9–3.3] | 47 | 10.5 [7.1–15.3] | 70 | 14.5 [10.8–19.3] |
| Married | 48 | 1.6 [1.1–2.2] | 179 | 5.6 [4.5–6.8] | 318 | 10.3 [8.7–12.1] |
| **Completed level of education** | | | | | | |
| No formal school | 25 | 0.7 [0.4–1.2] | 157 | 5.1 [4.2–6.3] | 282 | 9.6 [8.0–11.4] |
| Primary school | 11 | 1.6 [0.9–3.0] | 45 | 10.0 [6.9–14.2] | 67 | 14.7 [11.1–19.1] |
| Secondary or more | 24 | 11.8 [8.0–17.7] | 25 | 11.4 [7.2–17.6] | 40 | 18.0 [13.0–24.4] |
| **Profession** | | | | | | |
| Wage earner | 18 | 10.0 [6.3–15.6] | 14 | 6.0 [3.3–10.1] | 25 | 12.9 [8.5–19.0] |
| Self-employed | 26 | 0.9 [.06–1.5] | 127 | 5.0 [3.8–12.5] | 216 | 8.5 [6.9–10.5] |
| Jobless | 16 | 1.7 [1.0–3.0] | 85 | 9.5 [7.2–12.5] | 146 | 16.5 [13.2–20.5] |
| **Residence** | | | | | | |
| Urban | 37 | 4.4 [3.0–6.7] | 104 | 14.0 [10.4–18.6] | 156 | 21.6 [16.5–27.8] |
| Rural | 23 | 0.6 [0.4–1.1] | 123 | 3.6 [2.8–4.7] | 234 | 7.3 [6.1–8.7] |

ACE: American College of Endocrinology; AHA/NHLBI: American Heart Association and National Heart Lung and Blood Institute; JIS: Joint Interim Statement; CI: confidence interval; n = number of participants with metabolic syndrome.

hypertension [48]. The predominance of low HDL among the components of MetS among Africans has been documented [48, 49]. Sumner et al [50] have shown that a low HDL level is the most frequent MetS lipid pattern among African populations [50, 51]. The low HDL level among Africans represent a substantial and evolving cardiovascular risk. In this study, the high prevalence of low HDL might be due to high proportion of infection particularly in rural area. The low HDL is known to be associated with infection and systemic inflammatory response [52]. The pattern of dyslipidemia among African populations needs further investigations, which might be addressed by the Human Heredity and Health in Africa (H3Africa) initiative [53]. The high prevalence of dyslipidemia particularly low HDL among African populations means that we need to integrate the effective detection and treatment of dyslipidemia in primary care in Africa countries [49]. Abdominal obesity is the second most prevalent MetS component reported in our study. Abdominal obesity varied across the definition since different thresholds are used to define this component of MetS [54]. For descriptive analysis, we used the most recent thresholds for waist circumference measurement (JIS definition) to define abdominal obesity among the SSA adult population. Abdominal obesity is an important marker of insulin resistance but the relevant cut-off for African adults is still discussed [55, 56]. Due to the importance of MetS in the prevention of CVD, there is a crucial need to set up recommended normal range values of waist and hip circumference in Africans [57]. Hypertension is common among the adult population in SSA, nearly one third (30.8%) have high blood

**Table 6. Determinants of metabolic syndrome according to each of six definitions by sociodemographic characteristics.**

| Sociodemographic characteristics | WHO | NCEP-ATP III | IDF |
|---|---|---|---|
| | aPR [95%CI] | aPR [95%CI] | aPR [95%CI] |
| **Age group, years** | **P <0.001** | **P <0.001** | **P <0.001** |
| 25–34 | Ref | Ref. | Ref. |
| 35–44 | 2.49 [1.24–5.03] | 1.70 [1.11–2.61] | 1.67 [1.25–2.23] |
| 45–54 | 3.12 [1.51–6.42] | 2.44 [1.65–3.61] | 2.44 [1.89–3.16] |
| 55–64 | 4.12 [1.89–8.99] | 3.90 [2.51–6.04] | 3.30 [2.44–4.46] |
| **Gender** | **P = 0.75** | **P <0.001** | **P <0.001** |
| Women | Ref. | Ref. | Ref. |
| Men | 1.10 [0.61–2.01] | 0.35 [0.22–0.53] | 0.25 [0.17–0.37] |
| **Marital status** | **P = 0.59** | **P = 0.66** | **P = 0.52** |
| Single | Ref. | Ref. | Ref. |
| Married | 1.23 [0.60–2.49] | 1.08 [0.76–1.54] | 1.10 [0.83–1.45] |
| **Completed level of education** | **P = 0.06** | **P = 0.012** | **P = 0.71** |
| No formal school | Ref. | Ref. | Ref. |
| Primary school | 1.75 [0.89–3.47] | 1.76 [1.21–2.57] | 1.11 [0.82–1.50] |
| Secondary or more | 2.89 [1.15–7.28] | 1.38 [0.69–2.73] | 1.17 [0.72–1.89] |
| **Profession** | **P = 0.51** | **P = 0.96** | **P = 0.89** |
| Wage earner | Ref. | Ref. | Ref. |
| Self-employed | 1.14 [0.40–3.23] | 0.93 [0.44–1.98] | 1.02 [0.58–1.80] |
| Jobless | 1.62 [0.57–4.67] | 0.9 [0.41–1.98] | 1.09 [0.60–1.95] |
| **Residence** | **P = 0.043** | **P <0.001** | **P <0.001** |
| Urban | Ref. | Ref. | Ref. |
| Rural | 0.52 [0.29–0.95] | 0.35 [0.23–0.52] | 0.39 [0.29–0.52] |

The model was adjusted for all behavioral risk factors in Table 2; WHO: World Health Organization; NCEP-ATPIII: National Cholesterol Education Program Adult Treatment Panel III; IDF: International Diabetes Federation, CI: confidence interval, aPR: adjusted prevalence ratio.

pressure [58] and was the third highest prevalent component of MetS in our study. Elevated fasting blood glucose was the fourth most prevalent component of MetS. The prevalence of MetS decreased by 32% when excluded persons with diabetes [37]. MetS components are still poorly screened in Burkina Faso [44]. The most efficient preventive measures need to be implemented, including lifestyle modifications to address individual and population-wide cardiovascular disease risk in Burkina Faso, since the prevalence of some components of MetS is remarkably high.

The main determinants of MetS in our study were age and residence. We found that the prevalence of MetS was significantly higher in urban areas compared with rural areas regardless of the definition used. Similar findings have been reported in studies conducted in other west African countries including Nigeria [59], Benin [60] and Ghana [61]. Urbanization, associated with nutritional transition, and a sedentary lifestyle, has been linked to an increasing prevalence of MetS in SSA [62]. A higher prevalence of MetS in older age groups have been reported [45, 63]. The ongoing epidemiologic transition (with the aging of the population) contributes to increasing the prevalence of MetS in developing countries like Burkina Faso. In this study, we found out that for four out of six definitions of MetS (NCEP-ATP III, WHO, AACE, AHA/NHLBI), education was significantly associated with increased prevalence of MetS. The association between education level and MetS or diabetes is discussed in the literature. Indeed, Gatimu et al [64] have shown that the education was associated with increased odd of diabetes. Millogo et al [27] have not found this association. Nevertheless, high

**Table 7. Determinants of metabolic syndrome according to each of six definitions by sociodemographic characteristics.**

| Sociodemographic characteristics | AACE | AHA/NHLBI | JIS |
|---|---|---|---|
| | aPR [95% CI] | aPR [95% CI] | aPR [95% CI] |
| **Age group, years** | **P <0.001** | **P <0.001** | **P <0.001** |
| 25–34 | Ref. | Ref. | Ref. |
| 35–44 | 1.38 [0.70–2.71] | 1.88 [1.34–2.63] | 1.68 [1.31–2.17] |
| 45–54 | 2.21 [1.15–4.24] | 2.38 [1.72–3.27] | 2.33 [1.81–2.99] |
| 55–64 | 4.13 [1.99–8.56] | 3.79 [2.62–5.48] | 3.30 [2.49–4.36] |
| **Gender** | **P = 0.31** | **P <0.001** | **P <0.001** |
| Women | Ref. | Ref. | Ref. |
| Men | 0.74 [0.40–1.34] | 0.35 [0.24–0.52] | 0.36 [0.27–0.48] |
| **Marital status** | **P = 0.98** | **P = 0.50** | **P = 0.96** |
| Single | Ref. | Ref. | Ref. |
| Married | 1.01 [0.52–1.94] | 0.90 [0.65–1.24] | 1.01 [0.78–1.30] |
| **Completed level of education** | **P = 0.003** | **P = 0.07** | **P = 0.28** |
| No formal school | Ref. | Ref. | Ref. |
| Primary school | 2.00 [0.86–4.69] | 1.43 [1.03–1.99] | 1.20 [0.92–1.57] |
| Secondary or more | 4.06 [1.79–9.25] | 1.39 [0.78–2.49] | 1.28 [0.82–1.97] |
| **Profession** | **P = 0.10** | **P = 0.99** | **P = 0.55** |
| Wage earner | Ref. | Ref. | Ref. |
| Self-employed | 0.45 [0.21–0.94] | 1.13 [0.55–2.33] | 1.00 [0.59–1.71] |
| Jobless | 0.52 [0.22–1.2] | 1.31 [0.63–2.72] | 1.14 [0.66–1.97] |
| **Residence** | **P = 0.008** | **P <0.001** | **P <0.001** |
| Urban | Ref. | Ref. | Ref. |
| Rural | 0.36 [0.17–0.77] | 0.38 [0.26–0.54] | 0.46 [0.35–0.61] |

The model was adjusted for all behavioral risk factors in Table 2, ACE: American College of Endocrinology; AHA/NHLBI: American Heart Association and National Heart Lung and Blood Institute; JIS: Joint Interim Statement; CI: confidence interval, aPR: adjusted prevalence ratio.

education level is linked to the increasing adoption of new sedentary lifestyles and changes in dietary intake, which increase the risk of MetS and diabetes [64].

The normal range values of many components of MetS among the African population is still being discussed in the literature, and has motivated calls for a specific definition of MetS among this population [50, 51, 65]. However, it is known that individuals with a reduced number of MetS components have a lower risk of diabetes and cardiovascular diseases [66]. While waiting for the most accurate definition of MetS, substantial efforts are needed to screen and treat the clustering of components of MetS and to reverse the current rising trend of CVD in SSA including Burkina Faso.

## Strengths and limitations

The main limitation of this study was the unavailability of data on blood triglycerides, which were not measured during the survey due to the lack of adequate equipment that might underestimate the prevalence of MetS. Definitions of MetS had mainly focused on the western population and systematically included hypertriglyceridemia [45]. However most African people usually have normal triglyceride levels even so, the cardiovascular risk still higher among African population compared to the western population [50]. The data used in this study were collected since 2013, so the situation may have changed. There is a need to update this data to evaluate the temporal change of prevalence of MetS in Burkina Faso. The strengths of this

study were the use of different definitions of MetS to provide countrywide prevalence and a robust statistical approach to identify its determinants. This analysis was done in a country that is "starting" an epidemiological transition, which offers an opportunity to control the rising of MetS and NCDs early.

## Conclusion

The present study showed various prevalence of MetS in Burkina Faso depending on the definition. At the component level, we found that the prevalence of low HDL, obesity and hypertension were high in the adult population. We also reported that age and residence were the main determinants of MetS. The choose of appropriated definition to early detect and treat MetS in low-income countries such as Burkina Faso are crucial for achieving the sustainable goal of development by controlling the rising burden of CVD.

## Acknowledgments

The authors acknowledge Mady Ouédraogo, and Michel Bonkoungou for their contribution to data management. We would like to thank also Dr Sekou Samadoulougou for assisting with study design and statistical analyses.

## Author Contributions

**Conceptualization:** Fati Kirakoya-Samadoulougou.

**Data curation:** Kadari Cissé, Joel Dofinissery Bognini, Tiga David Kangoye.

**Formal analysis:** Kadari Cissé, Délwendé René Séverin Samadoulougou, Joel Dofinissery Bognini, Fati Kirakoya-Samadoulougou.

**Funding acquisition:** Fati Kirakoya-Samadoulougou.

**Methodology:** Fati Kirakoya-Samadoulougou.

**Project administration:** Fati Kirakoya-Samadoulougou.

**Resources:** Fati Kirakoya-Samadoulougou.

**Supervision:** Fati Kirakoya-Samadoulougou.

**Validation:** Fati Kirakoya-Samadoulougou.

**Writing – original draft:** Kadari Cissé, Délwendé René Séverin Samadoulougou.

**Writing – review & editing:** Kadari Cissé, Délwendé René Séverin Samadoulougou, Joel Dofinissery Bognini, Tiga David Kangoye, Fati Kirakoya-Samadoulougou.

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
