## [Decision Letter · Decision Letter 0]

26 Apr 2021

PONE-D-20-38221

Using the first nationwide survey on non-communicable disease risk factors and different definitions to evaluate the prevalence of metabolic syndrome in Burkina Faso

PLOS ONE

Dear Dr. Cisse,

Thank you for submitting your manuscript to PLOS ONE. After careful consideration, we feel that it has merit but does not fully meet PLOS ONE’s publication criteria as it currently stands. Therefore, we invite you to submit a revised version of the manuscript that addresses the points raised during the review process.

The manuscript has been evaluated by two reviewers, and their comments are available below.

The reviewers have raised a number of concerns that need attention. They request additional information on methodological aspects of the study, and modifications to the Limitations and Discussion sections. 

Could you please revise the manuscript to carefully address the concerns raised?

We look forward to receiving your revised manuscript.

Kind regards,

Carmen Melatti

Staff Editor

PLOS ONE

Journal Requirements:

5. Please include a caption for figure 1.

6. Please include a copy of Table 5b which you refer to in your text on page 14.

Reviewers' comments:

Reviewer's Responses to Questions

**Comments to the Author**

1. Is the manuscript technically sound, and do the data support the conclusions?

Reviewer #1: Yes

Reviewer #2: Yes

2. Has the statistical analysis been performed appropriately and rigorously? 

Reviewer #1: I Don't Know

Reviewer #2: Yes

3. Have the authors made all data underlying the findings in their manuscript fully available?

Reviewer #1: Yes

Reviewer #2: No

4. Is the manuscript presented in an intelligible fashion and written in standard English?

Reviewer #1: Yes

Reviewer #2: Yes

5. Review Comments to the Author

Reviewer #1: This is an important data for Burkina Faso.

Major comments:

The exclusion of TG criteria might contribute to the lower prev of MetS in this study. This should be addressed in the discussion section without extensive speculation on the possible relative unimportance of TG for Africans.

Minor comments:

The higher proportion of low HDL might also be related to the relatively higher infection in rural area. The author should address this in the discussion section.

To improve the clarity and simplicity of the table, the author might want to remove some of the details in the table.

For binomial data, it would be simpler if the author presented only one of the group instead of presenting both groups.

Eg only presenting % of men instead of presenting % of both men and women

only presenting % of subjects with hypetrtension instead of presenting % of subjects with hypertension and % of subjects without hypertension.

Reviewer #2: Thanks for asking me to review this important study on using the first nationwide survey on non-communicable disease risk factors and different definitions to evaluate the prevalence of metabolic syndrome in Burkina Faso. Studies of this nature are needed to inform policy decisions by funders.

In general, the article is very well written and I congratulate the authors for a job well done.

I however have a few minor observations

1. The data were collected almost 8 years ago. The data is based secondary data from a cross-sectional survey, the national WHO Steps survey 95 aiming to assess the risk factors for NCDs which was conducted in the 13 regions of Burkina Faso 96 from 26 September to 18 November 2013. With temporal changes and increasing western lifestyles, in most SSA countries, 8 years is ample time to not differences. The study is however such an important study for the people of Burkina Faso, that I don’t to recommend a rejection based on the age of the data. Instead, I will recommend an acknowledgement of this limitation, and in the discussion, the authors could mention the need for further a predictive modelling to account of temporal changes, or indeed, update the data if they can get funding.

2. Determinants of metabolic syndrome according to each of six definitions by sociodemographic characteristics: the discussion focused mainly on the characteristics that showed statistical significance. There needs to be some discussions around why, factors such as profession and educational attainment are not associated with metabolic syndrome. This is the finding in other similar studies from ssa. Gatimu et al. https://pubmed.ncbi.nlm.nih.gov/27871259/

3. Minor: Check the spelling on “Levesl” and “peoples”. In fact the 2 sentences need to be phrased better.

a. “The third step was to measure blood sugar levesl and blood cholesterol.

b. “type 2 diabetes [51]. However, in these conditions most African peoples usually have normal”

6. PLOS authors have the option to publish the peer review history of their article (what does this mean?). If published, this will include your full peer review and any attached files.

Reviewer #1: No

Reviewer #2: **Yes: **Dr Samuel Seidu

---

## [Author Response · Author response to Decision Letter 0]

5 Jul 2021

PONE-D-20-38221

Using the first nationwide survey on non-communicable disease risk factors and different definitions to evaluate the prevalence of metabolic syndrome in Burkina Faso

PLOS ONE

Dear Dr. Cisse,

Thank you for submitting your manuscript to PLOS ONE. After careful consideration, we feel that it has merit but does not fully meet PLOS ONE’s publication criteria as it currently stands. Therefore, we invite you to submit a revised version of the manuscript that addresses the points raised during the review process.

Thank you very much for the opportunity to revise our manuscript.

The manuscript has been evaluated by two reviewers, and their comments are available below.

We thank both reviewers for spending time to read our manuscript.

The reviewers have raised a number of concerns that need attention. They request additional information on methodological aspects of the study, and modifications to the Limitations and Discussion sections. 

Could you please revise the manuscript to carefully address the concerns raised?

We have addressed all of comments of academic editor and reviewers. Thank you.

 Please submit your revised manuscript by Jun 07 2021 11:59PM. If you will need more time than this to complete your revisions, please reply to this message or contact the journal office at plosone@plos.org. Please include the following items when submitting your revised manuscript:

We look forward to receiving your revised manuscript.

Kind regards,

Carmen Melatti

Staff Editor

PLOS ONE

Journal Requirements:

We have checked the references list. There is no article retracted. Thank you

 Thank you for this remark. We have added the following sentence: 

“The dataset of the STEPS survey that was used in this research is available at the Ministry of Health upon request to Bicaba Brice: bicaba_brico@yahoo.fr or Zoma Torez : torezo2000@yahoo.fr ). All survey materials are available on the WHO website (https://extranet.who.int/ncdsmicrodata/index.php/catalog)”. 

The majority of STEPS survey data are available upon request to the NCD Surveillance, Monitoring, and Reporting team (ncdmonitoring@who.int) or on the WHO website (https://extranet.who.int/ncdsmicrodata/index.php/catalog)”. the STEP data of Burkina Faso is not yet available on WHO website; however, it might obtain with ministry of health (bicaba_brico@yahoo.fr).

 Thank you

Thank you. We have moved the ethics statement to Methods section. (see line 161-167)

 5. Please include a caption for figure 1.

 Thank you for this remark. We have added a caption of Figure 1 in the manuscript. (see line 176)

6. Please include a copy of Table 5b which you refer to in your text on page 14.

Thank you for this remark. Table 5b was cited in the text however there was a mistake in the table number. we have corrected the number of Table in line 224

Reviewers' comments:

Reviewer's Responses to Questions

Comments to the Author

1. Is the manuscript technically sound, and do the data support the conclusions?

Reviewer #1: Yes

Reviewer #2: Yes

Thank you

2. Has the statistical analysis been performed appropriately and rigorously?

Reviewer #1: I Don't Know

Reviewer #2: Yes

Thank you

3. Have the authors made all data underlying the findings in their manuscript fully available?

Reviewer #1: Yes

Reviewer #2: No

There is no restriction to used the data but it is not available online. To obtain the data, researcher have to email the corresponding author of this manuscript or technicians of health ministry who have conducted the STEP survey ( Bicaba Brice: bicaba_brico@yahoo.fr or Zoma Torez : torezo2000@yahoo.fr ). Thank you. 

 4. Is the manuscript presented in an intelligible fashion and written in standard English?

Reviewer #1: Yes

Reviewer #2: Yes

Thank you

 5. Review Comments to the Author

Reviewer #1: This is an important data for Burkina Faso.

Thank you very much.

Major comments:

The exclusion of TG criteria might contribute to the lower prev of MetS in this study. This should be addressed in the discussion section without extensive speculation on the possible relative unimportance of TG for Africans.

We have reformulated the limitation section of the manuscript to delete the extensive speculation (see line 316-318). We addressed in the discussion section the lower prevalence of MetS in our study due to unconsidered TG in the definition of MetS. The part in the discussion section is formulated as follow: “The triglyceride level was not considered in the definition of MetS in our study since it was not collected during the STEP survey in Burkina Faso. This might contribute to the lower prevalence of MetS in this study.” (see line 242-244)

Minor comments:

The higher proportion of low HDL might also be related to the relatively higher infection in rural area. The author should address this in the discussion section.

Thank you for this suggestion. We have added a sentence to discuss the low HDL level. This sentence is “The low HDL level among Africans represent a substantial and evolving cardiovascular risk. In this study, the high prevalence of low HDL might be due to high proportion of infection particularly in rural area. The low HDL is known to be associated with infection and systemic inflammatory response [53].” (see line 268-271)

To improve the clarity and simplicity of the table, the author might want to remove some of the details in the table.

For binomial data, it would be simpler if the author presented only one of the group instead of presenting both groups.

Eg only presenting % of men instead of presenting % of both men and women

only presenting % of subjects with hypertension instead of presenting % of subjects with hypertension and % of subjects without hypertension.

Thank you for this suggestion. We have deleted the line in Table 3 accordingly (see Table 3 in line189). 

Reviewer #2: Thanks for asking me to review this important study on using the first nationwide survey on non-communicable disease risk factors and different definitions to evaluate the prevalence of metabolic syndrome in Burkina Faso. Studies of this nature are needed to inform policy decisions by funders.

In general, the article is very well written and I congratulate the authors for a job well done.

I however have a few minor observations

1. The data were collected almost 8 years ago. The data is based secondary data from a cross-sectional survey, the national WHO Steps survey 95 aiming to assess the risk factors for NCDs which was conducted in the 13 regions of Burkina Faso 96 from 26 September to 18 November 2013. With temporal changes and increasing western lifestyles, in most SSA countries, 8 years is ample time to not differences. The study is however such an important study for the people of Burkina Faso, that I don’t to recommend a rejection based on the age of the data. Instead, I will recommend an acknowledgement of this limitation, and in the discussion, the authors could mention the need for further a predictive modelling to account of temporal changes, or indeed, update the data if they can get funding.

Thank you for this suggestion. We have added a sentence to expression that in the discussion section. “The data used in this study were collected since 2013, so the situation may have changed. There is a need to update this data to evaluate the temporal change of prevalence of MetS in Burkina Faso.” (see line 323-325)

2. Determinants of metabolic syndrome according to each of six definitions by sociodemographic characteristics: the discussion focused mainly on the characteristics that showed statistical significance. There needs to be some discussions around why, factors such as profession and educational attainment are not associated with metabolic syndrome. This is the finding in other similar studies from ssa. Gatimu et al. https://pubmed.ncbi.nlm.nih.gov/27871259/

Thank you for this remark; we have added sentence to address this. See line 297 to 303 “In this study, we found out that for four out of six definitions of MetS (NCEP-ATP III, WHO, AACE, AHA/NHLBI), education was significantly associated with increased prevalence of MetS. The association between education level and MetS or diabetes is discussed in the literature. Indeed, Gatimu et al [64] have shown that the education was associated with increased odd of diabetes. Millogo et al [28] have not found out this association. Nevertheless, high education level is linked to the increasing adoption of new sedentary lifestyles and changes in dietary intake, which increase the risk of MetS and diabetes [64].” (see line 298-305)

3. Minor: Check the spelling on “Levesl” and “peoples”. In fact the 2 sentences need to be phrased better.

a. “The third step was to measure blood sugar levesl and blood cholesterol.

Thank you very much for this remark. we have corrected the spelling error. The sentence was rephrased as follow “At the third step blood sugar and blood cholesterol were measured”. (see line 117-118).

b. “type 2 diabetes [51]. However, in these conditions most African peoples usually have normal”

Thank you very much for this remark. we have corrected the spelling error. The sentence was rephrased as follow “However most African people usually have normal triglyceride levels even so, the cardiovascular risk still higher among African population compared to the western population [51].”. (see line 319-323).

6. PLOS authors have the option to publish the peer review history of their article (what does this mean?). If published, this will include your full peer review and any attached files.

Do you want your identity to be public for this peer review? For information about this choice, including consent withdrawal, please see our Privacy Policy.

Reviewer #1: No

Reviewer #2: Yes: Dr Samuel Seidu

Thank you.

We have corrected the Figure 1 accordingly.

---

## [Editor Report · Decision Letter 1]

21 Jul 2021

Using the first nationwide survey on non-communicable disease risk factors and different definitions to evaluate the prevalence of metabolic syndrome in Burkina Faso

PONE-D-20-38221R1

Dear Dr. Cisse,

We’re pleased to inform you that your manuscript has been judged scientifically suitable for publication and will be formally accepted for publication once it meets all outstanding technical requirements.

Kind regards,

Samuel Seidu, M.D, FRCP(Edin), FRCGP

Guest Editor

PLOS ONE
---

## [Editor Report · Acceptance letter]

29 Jul 2021

PONE-D-20-38221R1 

Using the first nationwide survey on non-communicable disease risk factors and different definitions to evaluate the prevalence of metabolic syndrome in Burkina Faso 

Dear Dr. Cissé:

I'm pleased to inform you that your manuscript has been deemed suitable for publication in PLOS ONE. Congratulations! Your manuscript is now with our production department. 

Kind regards, 

on behalf of

Dr. Samuel Seidu 

Guest Editor

PLOS ONE